# TEACHING TO TEACH BY STRUCTURED DARK KNOWLEDGE

## ABSTRACT

To educate hyper deep learners, *Curriculum Learnings* (CLs) require either human heuristic participation or self-deciding difficulties of training instances. These coaching manners are blind to the coherent structures among examples, categories and tasks, which are pregnant with more knowledgeable curriculum-routed teachers. In this paper, we propose a general methodology *Teaching to Teach* (T2T). T2T is facilitated by *Structured Dark Knowledge* (SDK) that constitutes a communication between structured knowledge prior and teaching strategies. On one hand, SDK adaptively extracts structured knowledge by selecting a training subset consistent with the previous teaching decisions. On the other hand, SDK teaches curriculum-agnostic teachers by transferring these knowledge to update their teaching policy. This virtuous cycle can be flexibly-deployed in most existing CL platforms and more importantly, very generic across various structured knowledge characteristics, e.g., diversity, complementarity and causality. We evaluate T2T across different learners, teachers and tasks, which significantly demonstrates that structured knowledge can be inherited by the teachers to further benefit learners' training.

## 1 INTRODUCTION

From an infant to a fully functional adult, human being requires years of highly advanced education. It purposively uses pedagogical instruments, demonstrates suitable examples and organizes targeted examinations, so as to reduce human being's time to equip with knowledge and skills. Drawing lessons from such social evolutionism, learning scientists proposed *teaching* Anderson et al. (1985);Goldman & Kearns (1995), a coined terminology that broadly refers to the frameworks and algorithms guiding better training qualities for complicated machine learners, e.g., networks and agents. One way to teach them is to demonstrate examples through following the leitmotiv "from-easy-to-hard", famous as *curriculum learning* (CL) Bengio et al. (2009). Specifically, Classical CLs (CCLs) Spitkovsky et al. (2010); Zaremba & Sutskever (2014) manage a *syllabus* (a dynamical training criteria) by ranking examples/tasks via increasing their difficulties from the perspective of human understanding [1]. Then the difficulty threshold continuously updates to tolerate harder examples and tasks (Fig.1. a.upper). Sometimes, CCLs inevitably entail human priors causing extra annotation and heuristically tuning, thus, are limited due to the algorithmic transferability. By contrast, Interactive CLs (ICLs) including SPL Kumar et al. (2010);SPCL Jiang et al. (2015);ACL Graves et al. (2017);TSCL Matiisen et al. (2017);MentorNet Jiang et al. (2017);sampling-based T2L Fan et al. (2018) and more, receive online training feedbacks as the difficulty signals to update their syllabuses. Their learner-teacher communication protocols reap the curriculum advantage without labor involvement (Fig.1. a.lower). So despite inconsistent performances across different learning scenarios Sachan & Xing (2016); Fan et al. (2018), ICLs maintain prevalent in the frontier of CL researches.

CCLs pay attention to human-explainable teaching while ICLs believe that only black-box learners can bring up a machine-suited curriculum. These cutting-edge studies indulging the argument about these pedagogical ordering styles, rarely realize that, most knowledge emerges as a natural macrocosm rather than a pile of isolated or artificially ordered pieces. For instance, a biological taxonomy consists of levels of creature species in a hierarchy; sentences and paragraphs imply logic rules of writing; a knowledge base bridges concepts embedded in undirected graphs. These structures integrate sporadic

---

[1] Example/task with less difficulty owns more priority in training, which presents as sampling with more frequency, or larger loss coefficient.

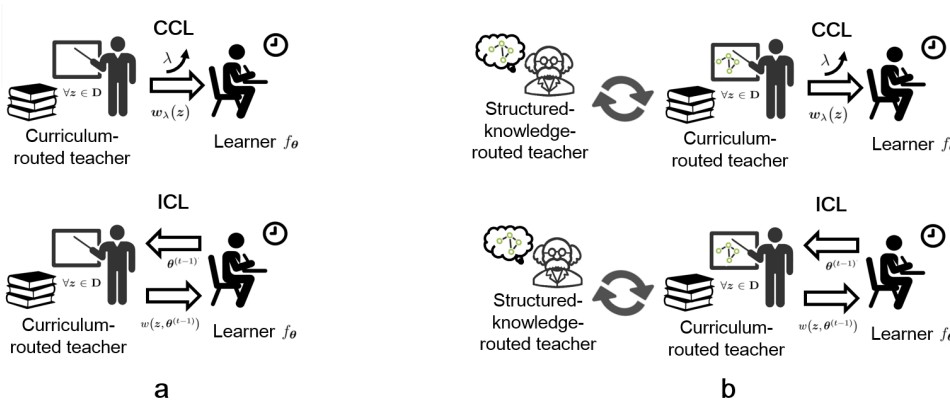

Figure 1: Overview of the iterative machine teaching procedures using CCL and ICL schemes on learner $f_\theta$. a). show the situations where CCLs and ICL normally perform. CCL increases difficulty threshold $\lambda$ to incorporete more training examples (training weight $w_\lambda(z)$ constantly increases); ICL builds up a teacher-learner communication protocol by training weight $w(z; \theta)$ to facilitate the alternative updating of teacher and learner. b). show our T2T where CCLs and ICL perform by interacting with a *structured dark knowledge cycle* (Section.3.2 ). Then CCL and ICL determine their teaching decisions based on not only the curricula but also the structured knowledge instructions.

pieces of knowledge, e.g., demonstrated examples, concepts (attributes and categories), multiple tasks, to reflect cognitive association characteristics, e.g., ambiguity, diversity, complementarity, causality, etc. It promises across-the-aboard education quality and has already grabbed a great amount of attentions in the field of pedagogical psychology Kirkpatrick & Epstein (1992). Regretfully, retrospect to massive literatures related to CLs in details and we find that, seldom were proposed under this consideration. A few of studies directly treated the case as an evolving training subset selection with diversity, based upon self-paced learning Jiang et al. (2014) Zhou & Bilmes (2018) or some tricks for a concrete problem Sachan & Xing (2016). They are situated under the difficulty-specific background and absent for generalization. Besides, their selected sets are directly used to optimize the learner, which implies the deterministic binary weights of training examples. It performs inferior compared with the other CL strategies promoting soft weighting or stochastic sampling technique.

Just as American novelist Ralph Ellison said " Education is all a matter of building bridges ", to bridge pieces of knowledge therefore rectify curriculum-based machine education, is what our work chases after. Note that, rather than crafting a specific CL algorithm or framework, we prefer ***teaching to teach*** (T2T), namely, distilling the substructure to dig out the possible coherence of training instances, e.g., examples, categories and tasks, then teach curriculum-routed teachers from existing CLs. As illustrated by the comic in Fig.1 .b, our methodology extends CCLs and ICLs by a teacher of teacher mastering structured knowledge, which is transferred to curriculum-routed teachers within a virtuous cycle: On one hand, this teacher of teacher "comprehends" the previous CCL/ICL teaching decisions, thus adaptively selects a knowledge-based substructure to "instruct" the curriculum-routed teachers. On the other hand, the curriculum-routed teachers merge the curricula with their updated structured knowledge to iteratively polish their teaching strategies toward learners.

More specifically, let's go through the technical discussion roadmap of this paper. In Section.3.1, we first revisit CCL and ICL approaches and frame them into the background of progressive reweighting learning. Then we observe that, the weight of each training instance inferred by CCL/ICL is valued in $[0, 1]$. In a stochastic sampling circumstance, they also represent the instance-selection probabilities, thus, the inference can be regulated by KL divergence, similar to *transferring dark knowledge* Hinton et al. (2015) by matching their activation outputs. But how to construct the structured knowledge to properly regulate the probabilistic weight inferences ? In Section.3.2, we introduce a set function as our structured knowledge prior, namely, the teacher of teacher shown in Fig.1 .b. The set function could be submodular Fujishige (2005) or just preserves submodular-like properties Das & Kempe (2011) Zhou & Spanos (2016b) (More specification refers to our Appendix.B). Our set function cooperates with the previous CL teaching decisions by matching their outputs, thus selects a subset of training instances to adaptively extract the structured knowledge. When curriculum-routed teaching strategy updates, this activation matching plays a role to transfer the structured knowledge to infer the curriculum teaching strategy. This cycle can be flexibly deployed in most existing CL strategies then incorporate structured knowledge to enhance their teaching performances.

Compared with previous work about CLs, our methodology embodies three apparent virtues:

- **Generality for structured knowledge.** Our methodology employs a generic set function as the structured knowledge prior to perform a constrained subset selection, where the structured dark knowledge only performs as a modular function. It is in harmony with diverse forms of subset selection, thus, refers to various structured knowledge among data.

- **Flexibility for curricula.** Our methodology could be flexibly deployed on all curriculum-based teaching strategies driven by, e.g., *models, algorithms, preset rules, even choices from human beings*, as long as they obey the generalized CCL and ICL formulas in SubSection.3.1.

- **Simplicity for implementation.** Our methodology connects teaching and teaching to teach with only a structured dark knowledge term conceptually simple for implementation.

Finally, we verify our methodology in three scenarios: classification, domain adaptation and sequence learning. We import the structured knowledge priors about diversity, complementarity and causality to their experimental setups. Empirical studies across diverse learners and teachers show that structured dark knowledge can be acquired by the teachers and helpful for them to educate the learners better.

## 2 RELATED LITERATURES

Teaching has gradually become an attractive AI research direction Khan et al. (2011);Zhu (2015);Zhu (2013). Our work keeps cohesive with two thriving trends about how to teach.

### 2.1 TEACHING TO LEARN WITH CURRICULUM

Plenty of researches concerned the learning principle *starting small* Elman (1993). Bengio et al. (2009) suggested that a series of training criterion by increasing the sample-based learning difficulties are able to accelerate the trainings or improve the performances of networks. This cognition-steered ideology was interpreted as *curriculum learning* (CL). Though born as a heuristic for practitioners, CL attracts increasing interests of theoretician in explaining, e.g., *extreme strategy* in teaching dimension Khan et al. (2011), relationship with importance sampling Katharopoulos & Fleuret (2018). Under some circumstance, the CLs using *ideal difficulty score* Weinshall & Cohen (2018) are proved to boost linear regression learner's convergence rate. Doubtlessly, CLs inspired a line of subsequent deep-model-based AI investigations about how to imitate human behaviors, e.g., BCD number calculation Zaremba & Sutskever (2014), game shooting from the first-person aspect Wu & Tian (2016), etc, and especially suit robotic control Sanger (1994) Florensa et al. (2017). Since these CLs are human-designed, we call them classical CLs (CCLs) in this paper.

Syllabuses in CCLs proceed in accordance with predefined schemes. For instance, a robotic arm is supposed to acquire the grasping motions from elementary to complex. By contrast, interactive curriculum learnings (ICLs) Graves et al. (2017);Kumar et al. (2010) prefer the syllabuses in adaptive dynamics, namely, self-refine the difficulties to keep consistent with the learner's training feedbacks. This learner-oriented manner appeals to the co-evolution of the learner and the teacher to execute curriculum. The pioneering researches track back to *Self-paced learning* (SPL) Kumar et al. (2010) and its variants Jiang et al. (2015);Jiang et al. (2014);Zhou & Bilmes (2018). They directly take current-step training losses as the difficulty feedbacks, then, are programmed to teach by reweighting losses in the constraints of scheme functions. Recently, using various kinds of learning progress signals Houthooft et al. (2016), ACLGraves et al. (2017), STCLMatiisen et al. (2017) and OACL Doan et al. (2018) apply bandit-based sampling Auer et al. (2002) to teach deep learners and GANs. Attractive progresses of ICL even employ a network to guide network training. For example, MentorNet Jiang et al. (2017), ScreenNet Kim & Choi (2018) focus on learnable reweighting schemes; Ho et al. (2016);Milli et al. (2017) Fan et al. (2018) employ agents to select training samples. The latters are known as sampling-based teaching, while also belong to ICL in a broad sense.

### 2.2 TEACHING TO LEARN WITH DARK KNOWLEDGE

The development of dark knowledge originates from the mockup experiments about hyper energy physics, where the deep networks were leveraged to search for dark matter from the particle collider synthesis Sadowski et al. (2015). Hinton et al. (2015) elaborated the method as *knowledge distillation*

(KD), namely, seeking to transfer *dark knowledge*, the soft predictions containing not only correct but also meaty ambiguous wrong informations, from a sophisticated teacher to fresh student networks by matching their output activations. The primitive goal of dark knowledge is to compress a complicated model into a lightweight one while retaining comparative performance. There could be multiple teachers for one student Rusu et al. (2015);Ruder et al. (2016), where the student is supposed to inherit all the teachers' capabilities. Recent researches discovered its potential to boost student network's performance Furlanello et al. (2018). It explains the successes of its wide applications in the other areas Shin et al. (2017);Papernot et al. (2016);Lopez-Paz et al. (2015).

Dark knowledge is broadly deemed as the gift from deep learning, whereas also surprisingly successful by hiring a teacher from the other ML areas. Hu et al. (2016) designed a rule-based teaching with soft first-order logic, then the student nets are iteratively updated to obey these principles. In Bayesian dark knowledge Korattikara et al. (2015), a Monte Carlo posterior predictive density (e.g., stochastic gradient Lagevin dynamic) educates a student to endow a probabilistic model reaping the benefit of Bayesian tools yet only costing the same run time as those plug-in methods.

## 3 CURRICULUM MARRIES DARK KNOWLEDGE

Let's begin with a typical supervised learning setting where $\mathcal{D} = \{(\boldsymbol{x}_i, \boldsymbol{y}_i)\}_{i=1}^{|\mathcal{D}|}$ [2] denotes training set and $f_{\boldsymbol{\theta}}(\cdot)$ denote a machine learner w.r.t. $\boldsymbol{\theta}$. To include all situations in magnificent literatures about CL, we reconsider $\mathcal{D}$ as $\mathbb{D} \subseteq \{\boldsymbol{z} \in 2^{|\mathcal{D}|}, |\boldsymbol{z}| = m\}$ and treats $\boldsymbol{z}$ as a single training instance. Correspondingly, $\mathcal{L}(\boldsymbol{z}; \theta)$ denotes a empirical risk based on a surrogate loss over $\forall \boldsymbol{z} \in \mathbb{D}$. In this case, $\boldsymbol{z}$ could be a training example ($m = 1$), or a group of samples belonging to the same class or task with the same size $m$. The primary objective is proposed as

$$\min_{\boldsymbol{\theta}} \mathcal{L}_f(\boldsymbol{\theta}) := \mathbb{E}_{\boldsymbol{z} \sim P(\boldsymbol{z})} \mathcal{L}(\boldsymbol{z}; \boldsymbol{\theta}). \tag{1}$$

where $P(\boldsymbol{z})$ denotes the base training distribution over $\mathbb{D}$ and usually is uniform.

### 3.1 CURRICULA GENERALLY REVISITED

CLs advocate the teaching through reconfiguring the training prior $P(\boldsymbol{z})$ based upon the instances' difficulties: Less difficult instance earns larger weight or sampling frequency during training. Distinguished from the update rules, they basically are categorized into two branches: CCL and ICL.

**Classical curriculum learning (CCL).** A human prior mapping $w_\lambda(\boldsymbol{z}) \in [0, 1]$ is employed by CCL to measure difficulty: Given a scalar $\lambda > 0$ per iteration and $\forall \boldsymbol{z}_1, \boldsymbol{z}_2 \in \mathbb{D}$, $\boldsymbol{z}_1$ maintains harder than $\boldsymbol{z}_2$ iff $w_\lambda(\boldsymbol{z}_1) \leq w_\lambda(\boldsymbol{z}_2)$. Specify $Q(\boldsymbol{z}; \lambda) \propto w_\lambda(\boldsymbol{z})P(\boldsymbol{z})$ then Eq.1 evolves to

$$\min_{\boldsymbol{\theta}} \mathcal{L}_{\text{CCL}}(\boldsymbol{\theta}; \lambda) := \mathbb{E}_{\boldsymbol{z} \sim Q(\boldsymbol{z}; \lambda)} \mathcal{L}(\boldsymbol{z}; \boldsymbol{\theta}). \tag{2}$$

. The objective performs as a series of training criterion as $\lambda$ iteratively increases. $Q(\boldsymbol{z}; \lambda)$ indicates the dynamical training distribution of samples. It is emphasized that, the increase of $\lambda$ rigorously does not change the order of their weights. $\frac{\partial w_\lambda \boldsymbol{z}}{\partial \lambda} > 0$ promises as CCL proceeds, training samples are ultimately accepted in a fair proportion. Due to the definition about training instance $\boldsymbol{z}$, existing supervised learning frameworks that involve manual curriculum can be concluded as Eq.2 .

**Interactive curriculum learning (ICL).** CCL curates *hand-crafted difficulty* and update the training syllabus by increasing $\lambda$. In contrast, interactive curriculum learning (ICLs) receive training feedbacks to estimate the *learner-oriented difficulty*. Assume $f_{\boldsymbol{\theta}^{(t-1)}}$ as the learner previously trained in the iteration $t-1$, then the current weight for $\boldsymbol{z}$ is written as $w(\boldsymbol{z}; \boldsymbol{\theta}^{(t-1)})$ that represents the past training feedback. Using $Q(\boldsymbol{z}; \boldsymbol{\theta}^{(t-1)}) \propto w(\boldsymbol{z}; \boldsymbol{\theta}^{(t-1)})P(\boldsymbol{z})$ ICL presents as

$$\min_{\boldsymbol{\theta}^{(t)}} \mathcal{L}_{\text{ICL}}(\boldsymbol{\theta}^{(t)}; \boldsymbol{\theta}^{(t-1)}) := \mathbb{E}_{\boldsymbol{z} \sim Q(\boldsymbol{z}; \boldsymbol{\theta}^{(t-1)})} \mathcal{L}(\boldsymbol{z}; \boldsymbol{\theta}^{(t)}). \tag{3}$$

Distinguished with the difficulties pre-planned by $w_\lambda(\boldsymbol{z})$, Eq.3 develops the training criterion with $w(\boldsymbol{z}, \boldsymbol{\theta}^{(t-1)})$ where the difficulties are hidden. Conventionally, a well-performed training example in

---

[2] $\boldsymbol{x}_i \in \mathcal{X}$ over data space in $\mathbb{R}^d$, and $\boldsymbol{y}_i \in \mathcal{Y}$ over label space in $\mathbb{N}_+$ for classification or $\mathbb{R}$ for regression. For multi-task and structured learning, $\mathcal{Y}$ turn to be high-dimensional.

the previous iter is considered easy and beneficial to train the current model. It obtains a larger weight or higher stochastic sampling frequency in this training phase. Realize that, ICLs (Eq.3) should be implemented by specifying $w(z, \theta^{(t-1)})$. Most ICLs implicitly contain their weight functions. In our Appendix.A, we elaborates them and show how to explicitly relate their strategy to $w(z, \theta^{(t-1)})$.

**Probabilistic shadows of curriculum-routed teachers.** The curriculum-routed teachers of CCLs and ICLs might be *explicit* (updated by differentiable scheme functions Jiang et al. (2017)), but more commonly are *implicit* (updated by an algorithm, a rule, an isolated agent or model). We observe that, their strategies take effects in training via $w_\lambda(z)$ or $w(z, \theta^{(t-1)})$. These weights are located in $[0, 1]$, then in a stochastic manner, approximately equal to the probabilities to select $\forall z \in \mathbb{D}$. Precisely, we use $P_w$ to reconfigure the weights into a set of selection probabilities,

$$\forall z_i \in \mathbb{D}, \; P_w(g_i) = \begin{cases} w_i(1 - 2\epsilon) + \epsilon & if \; g_i = 1 \\ 1 - \epsilon - w_i(1 - 2\epsilon) & if \; g_i = 0 \end{cases} \tag{4}$$

where $g_i$ denotes a binary random variable representing a selection flag of $z_i$. $w_i$ unifies $w_\lambda(z_i)$ and $w(z_i, \theta^{(t-1)})$ for simplicity. $0 < \epsilon << 1$ prevents the arithmetic pathological problem when applying $\log$ operator. In fact, given $\epsilon \to 0$, it preserves $P_w(z_i) \to w_i$.

The above analysis implicates two perspectives. First, transferring knowledge to regulate the weight update, equally acts on teaching the curriculum-routed teachers. Second, due to the probabilistic view of $w_\lambda(z)$ and $w(z; \theta^{t-1})$, this knowledge could be transferred by matching the activations between the teaching decision $P_w(g_i)$ and some other selection probabilistic decision of structured knowledge. The second point of view is very similar to knowledge distillations (see more comparison in Table.1 ).

## 3.2 Structured Dark Knowledge Cycle

For teaching to teach, we develop a new way of probability matching termed *Structured Dark Knowledge*. On one hand, it helps

Table 1: Comparison of dark knowledge (DK) approaches.

| | DK approaches | | |
| --- | --- | --- | --- |
| | Standard DK | Bayesian DK | Structured DK (Ours) |
| Teacher | Networks | MC Posterior Density | Training Subset |
| Student | Networks | Networks | CL Strategies (can be networks) |
| Update | Fixed Teacher | Alternative Update | Alternative Update |

to discover structured knowledge, namely, selects a subset $S \subseteq \mathbb{D}$ that illustrates the structured coherence in $\mathbb{D}$ and at the same time, keeps the dark-knowledge-driven consistency with the teaching decisions $P_w(\cdot)$. On the other hand, it has been re-applied to transfer structured knowledge into the teacher, so as to influence the teaching decision update. This cycle iteratively performs to guide a curriculum-routed teacher absorbing structured knowledge to teach a learner.

**Structured knowledge prior.** There is no such thing as free lunch and the structured knowledge is no exception. To bring the external knowledge into teaching, we should first import a set function $F(S)$ ($\forall S \subset \mathbb{D}$) as a carrier to embed structured knowledge prior, because a set is the basis to present magnificent kind of structure representations Anderson et al. (1985).

**Structured dark knowledge**. Provided a set function $F(\cdot)$ over $\mathbb{D}$ as our prior constraint, we expect to select a subset $S \subseteq \mathbb{D}$ based upon not only the structured knowledge but also the consistency with the teaching decision $P_w(\cdot)$. The consistency is constructed by a knowledge distillation term matching their selection decisions in probability. To achieve this goal, we revisit $\forall S \subseteq \mathbb{D}$ from a probabilistic view like $w_i$:

$$if \; z_i \in S, \; P_F(g_i; S) = \begin{cases} 1 - \epsilon & g_i = 1 \\ \epsilon & g_i = 0 \end{cases}, \; if \; z_i \in \mathbb{D}/S, \; P_F(g_i; S) = \begin{cases} \epsilon & g_i = 1 \\ 1 - \epsilon & g_i = 0 \end{cases} \tag{5}$$

Note that, $P_F(\cdot; S)$ means *the probability under the background of subset $S$, but not conditional density*. Obviously, the density set $\{P_F(g_i; S)\}_{i=1}^{|\mathbb{D}|}$ refers to the subset selection result $S$ if $\epsilon \to 0$.

Then we propose a *structured dark knowledge* term $D_{\text{SDK}}$ to correlate $P_F(g_i; S)$ and $P_w(g_i)$:

$$D_{\text{SDK}}(z_i, w; S) = D_{\text{KL}}(P_F(g_i; S) | P_w(g_i))$$

$$= \begin{cases} -(1 - \epsilon) \log \frac{w_i(1-2\epsilon)+\epsilon}{1-\epsilon} - \epsilon \log \frac{(1-\epsilon)-w_i(1-2\epsilon)}{\epsilon} & if \; z_i \in S \\ -\epsilon \log \frac{w_i(1-2\epsilon)+\epsilon}{\epsilon} - (1 - \epsilon) \log \frac{(1-\epsilon)-w_i(1-2\epsilon)}{1-\epsilon} & if \; z_i \in \mathbb{D}/S \end{cases} \tag{6}$$

where $D_{\mathrm{KL}}$ indicates the Kullback–Leibler (KL) divergence.

Structured dark knowledge maintains two interpretations replaying in a cycle. In the context of subset selection, it presents as the consistency measure on each instance between the specified structured knowledge (namely, subset $S$) and the previous teaching decision $P_w$. It leads to their instance-level disagreement. Hence we are able to define the knowledge-based subset selection as

$$\max_{S \subseteq \mathbb{D}, |S| \leq k} \mathcal{J}(S) = F(S) + \beta \sum_{\boldsymbol{z}_i \in S} \left[ 1 - \frac{D_{\mathrm{SDK}}(\boldsymbol{z}_i, w; S)}{C} \right] \tag{7}$$

where $\gamma$ is a balance factor and $|S| \leq k$ is a cardinality constraint. $C = -(1 - 2\epsilon) \log \frac{\epsilon}{1-\epsilon}$ upperbounds $D_{\mathrm{SDK}}(\boldsymbol{z}_i, w|S)$ when $\boldsymbol{z}_i \in S$, thus, promises the second term always greater than $0$ [3]. Provided $\boldsymbol{z}_i$ and $\boldsymbol{z}_i$ performing close in the structured knowledge prior ($F(S \cup \{\boldsymbol{z}_i\}/\{\boldsymbol{z}_j\}) \approx F(S)$), a smaller $D_{\mathrm{SDK}}(\boldsymbol{z}_i, w|S)$ indicates a good matching between the knowledge and curriculum on instance $\boldsymbol{z}_i$, which is more preferable to be a candidate.

---

**Algorithm 1** T2T by SDK: General Algorithm

---
1: **for** each epoch **do**
2:     $S \leftarrow \arg \max_{S \subseteq \mathbb{D}, |S| \leq k} \mathcal{J}(S)$
3:     **for** each mini-batch **do**
4:         **if** $\boldsymbol{w} := \boldsymbol{w}_\phi$ **then**
5:             $\phi \leftarrow \arg \min_{\phi} \mathcal{O}(\phi; S), \boldsymbol{w} \leftarrow \boldsymbol{w}_\phi$
6:         **else**
7:             $\hat{\boldsymbol{w}} \leftarrow \arg \min_{\hat{\boldsymbol{w}}} \mathcal{O}(\hat{\boldsymbol{w}}; S), \boldsymbol{w} \leftarrow \hat{\boldsymbol{w}}$
8:         **end if**
9:         Formulate $\min_{\boldsymbol{\theta}} \mathcal{L}_{\mathrm{CCL}}(\boldsymbol{\theta}; \lambda)$ or $\min_{\boldsymbol{\theta}^{(t)}} \mathcal{L}_{\mathrm{ICL}}(\boldsymbol{\theta}^{(t)}; \boldsymbol{\theta}^{(t-1)})$ by $\boldsymbol{w}$. Solve it to obtain $\boldsymbol{\theta}$.
10:     **end for**
11: **end for**
12: **return** $\boldsymbol{\theta}^* = \boldsymbol{\theta}$.

---

The above objective is mostly NP-hard without any specification of the structured knowledge prior $F$. Yet under some situations (e.g., $F$ obeys submodularity), it can be solved with a provable approximate ratio. Due to the space limit, we are going to discuss more concrete definitions and examples of $F$ in our Appendix.B, which embed different structured knowledge indicating, e.g., diversity, complementarity and causality. The algorithms corresponding to those characteristics are also provided to solve Eq.7 .

Suppose the structured knowledge has been distilled by $S$, we turn to transfer them into our curriculum-routed teaching strategy. Under this context, $D_{\mathrm{SDK}}(\boldsymbol{z}_i|S)$ become a constraint to update $P_w$.

**Structured knowledge transfer.** The main challenge is to transfer the structured knowledge into *a teacher* instead of *a learner*. A learner is usually a parameterized model or agent updated by a differentiable learning process, which our structured dark knowledge can directly join to regulate. However, due to an extensive array of studies about CL, their teachers can be an algorithm, a fix update rule even directly coming from intentional choices of human being. One outlet is to infer a new strategy to balance the decisions of curriculum and the structured knowledge. Employing $\hat{w}_i = \hat{w}(z_i)$ ($\forall z_i \in \mathbb{D}$) to present a new balance strategy. We propose the objective as

$$\min_{\hat{\boldsymbol{w}} \in [0,1]^{|\mathbb{D}|}} \mathcal{O}(\hat{\boldsymbol{w}}; S) = \sum_{\boldsymbol{z}_i \in \mathbb{D}} ||w_i - \hat{w}_i||_2^2 + \gamma D_{\mathrm{SDK}}(\boldsymbol{z}_i, \hat{w}; S) \tag{8}$$

where the weights are inferred by gradient decent. However, the most desirable case is that the teacher is also a learner and optimized by the loss function with $f_\theta$'s training feedback. Hence we are able to transfer the knowledge by posing $D_{\mathrm{SDK}}(\boldsymbol{z}_i|S)$ on the teacher learning objective. Given $w_\phi$ as this learnable teacher with parameter $\phi$, the objective presents

$$\min_{\boldsymbol{\phi}^{(t)}} \mathcal{O}(\boldsymbol{\phi}^{(t)}; S) = \mathcal{M}(w_{\boldsymbol{\phi}^{(t)}}, f_{\theta^{(t-1)}}) + \gamma \sum_{\boldsymbol{z}_i \in \mathbb{D}} D_{\mathrm{SDK}}(\boldsymbol{z}_i, w_{\boldsymbol{\phi}^{(t)}}; S) \tag{9}$$

where $\mathcal{M}(w_{\boldsymbol{\phi}^{(t)}}, f_{\theta^{(t-1)}})$ indicates the original teacher objective, where the current teacher is updated by the feedbacks from the previous learner. If the teacher is a RL agent , $\mathcal{M}(w^{(t)}, f_{\theta^{(t-1)}})$ is the negative of the value function and trained under the MDP setting.

---

[3]The positive subset selection objective is quite welcome in many combinatorial optimization setup.

## 4    EXPERIMENTS

In this section, we evaluate our methodology by comprehensive empirical studies. They include most existing CL approaches across diverse tasks, from transfer learning to computational finance.

### 4.1    BASIC EXPERIMENTAL SETTING.

Before introducing our empirical studies, let's specify some common experimental details. For the setting of our structured dark knowledge (SDK), $\epsilon = 0.04$ if the targeted CL applies reweighting strategy (e.g., SPL family); and set $\epsilon = 0.1$ if they apply stochastic sampling schemes. It provides more exploration freedom for the sampling-based CL approaches. For the cardinality constraint in Eq.7, $k$ starts from selecting $15\%$ instances to construct $S$, then after per epoch, linearly increases till $k = |\mathbb{D}|$. The duration is decided by the speed of the original learner's convergence. The motivation is to promise, if $F$ monotonically increases, all the samples will be finally included.

#### 4.1.1    CURRICULA AS APPRENTICES.

- **Preset Curriculum (PC).** To thoroughly understand the benefit of structured knowledge, our experiments should include CCLs. But traditionally, their curricula are manually-planned, task-oriented and worst of all, subjective across people. For a fair comparison, we follow the recent study Weinshall & Cohen (2018) and train a "prophet" network (higher capacity than the learner, we apply ResNet101 He et al. (2016) in our classification experiment), then rank the training data by increasing their losses on this well-trained network. Then we use this schedule to perform stochastic sampling as the preset curriculum (PC) in CCL.

- **SPL** (Eq.12), **SPCL** (mixture) (Eq.15). SPL and its variants are the elementary ICL methods. According to their teaching principles, the training instances with larger losses are assigned smaller weights, then a threshold grows to tolerate the large losses so that eventually, all training losses are assigned equal coefficients. For the robustness in a stochastic manner, we follow the routine in Jiang et al. (2014) : For the data in each mini-batch, SPL Kumar et al. (2010) and SPCL (mixture) Jiang et al. (2015) infer their weights, thus, we filter out top $K$ weight-smallest samples. $K$ starts from a quarter of a mini-batch, then it linearly decreases to $0$ after $E$ epochs.

- **ACL** (Eq.16). ACL belongs to the sub-branch of ICLs using a stochastic bandit to sample multiple tasks and construct mini-batches online. Each slot in the bandit refers to a task, and the reward comes from diverse learning progress signals. We use *self-prediction gain* signal (Eq.21 ) that shows an unbias progress estimation during the stochastic learning. We directly borrow the hyperparameter setup in Graves et al. (2017), which has already been good enough to present ACL's superiority. ACL is only applied in multi-task learning.

In our structured dark knowledge transfer, the weights of the above CL methods are refined by Eq.8, then the newly inferred weights replace the originals to execute their CL strategies.

- **Learning to teach (L2T)** (Eq.24) Sampling-based L2T Fan et al. (2018) specifies a RL-based teacher agent to select training instances. Different with the above methods, L2T suits to verify whether structured dark knowledge can benefit the learnable CLs 9. In our setup, $\gamma D_{\text{SDK}}(z_i, w_{\phi^{(t)}}; S)$ constrains L2T's binary policy decision of its teacher network (Under this context, $w_{\phi^{(t)}}$ denotes the teacher policy network with binary classification).

More implementations about PC and L2T are deferred in Appendix.C.

#### 4.1.2    ORACLE STRUCTURED KNOWLEDGE.

Our evaluation is conducted on several widely-adopted standard benchmarks in ML researches. They do not naturally provide structured knowledge among data. To built up this knowledge, we accept the "prophet" setting in PC. Specifically for the first experiment, we train a ResNet101 on the targeted benchmark and extract the training data before the last activation. These features marked as $f^*$ lying on an oracle semantic space, are used to measure the similarities among training data. For our second experiment, we follow a similar routine on the source and target domains respectively.

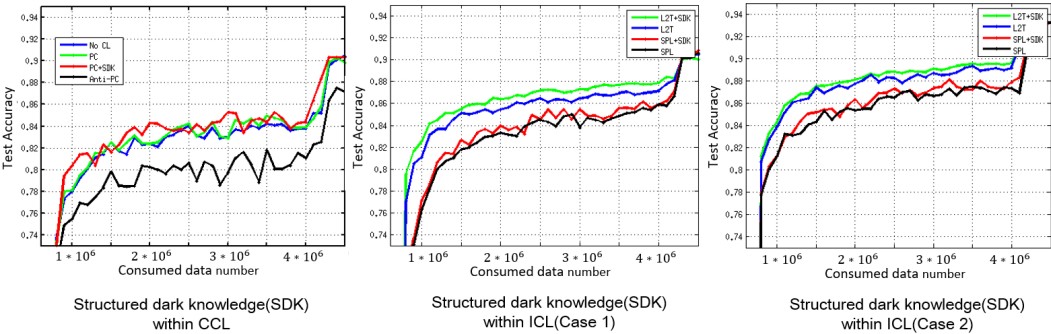

Figure 2: Overview of three comparison settings for CLs. The first evaluates PC+SDK by comparing the original PC, random teaching (No CL) and anti-curriculum (Anti-PC, weights inversely proceeds accorinding to PC). The second and third evaluate SDKs acting on ICL methods, e.g., SPL and L2T.

The structured knowledge priors $F$ in our evaluation include *Concept*(Eq.38), *BP function*(Eq.39) and *Granger Causality* (GC). They present the characteristics about diversity, complementarity and causality. Their constructions partly (e.g., some of Concept and BP function) base on similarities among data, which are obtained by the oracle knowledge setup.

## 4.2 TINY IMAGE CLASSIFICATION

Our first experiment is conducted on CIFAR-10 Krizhevsky & Hinton (2009). We employ ResNet-32 as the backbone learners. Momentum-SGD Sutskever et al. (2013) is the solver during training.

**Curricula setup.** PC, SPL and L2T are considered in this empirical study. The setups of PC, SPL follows SubSection.4.1.1. L2T performs the both testing settings in the original paper. In the first setting, we first train a three-layered MLP teacher network by interacting with a student network, then fix the teacher's parameter to retrain a student from scratch. Note that, the first and second training sets are split from a same dataset without an overlap. Thus, we perform L2T by only half of the training set in CIFAR10. For the second setting, we train the same teacher by teaching a simple LeNet student on MNIST. Then we directly use it to instruct the ResNet32 learning on the entire training set of CIFAR10. If our T2T is applied, both teacher trainings will be regulated by our SDK.

**Knowledge setup.** Diversity is the investigated characteristic: we employ the submodular-based *Concept* (Eq.38) as our structured knowledge prior and its specifications are shown in Appendix.C. $\max_{S \subseteq \mathbb{D}, |S| \leq k} \mathcal{J}(S)$ is solved by Local Search (Algorithm.2 ).

EMPIRICAL RESULTS AND IMPLICATIONS.

**Evaluation.** PC, SPL and L2T can be viewed as sampling-based curriculum strategies. To illustrate the data-efficiencies they bring about, we evaluate the accuracies of their underlying learners according to their growing consumption of training data. To effectively reveal this case, we borrow the principle in Fan et al. (2018): During their teaching processes, the learners will not be updated till $M$ untrained, thus, to accumulate the selected instances. It promises the convergence speed relying on the quality of selected instances, which provides a good observation to evaluate curriculum strategies, and their chemical reactions with structured dark knowledge. Fig.3.1 illustrate three circumstances: The sub-figures

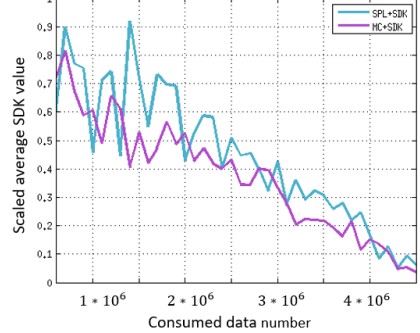

Figure 3: The scaled averaged SDK values of SPL and PC as the training proceeds.

from left to right, indicate the evaluations under CCL, ICL(the first test setting in L2T), ICL(the first test setting in L2T). SDKs basically perform envelopes that improve the targeted PC, SPLs and L2Ts. More detailedly, compared with CL, ICL obtain mild improvement margins, and L2Ts obtain more

benefits than SPLs. It probably implies that, the way of using SDK by Eq.9 performs more harmony with the learnable teaching methods.

**Finding the Schrodinger's cat of structured dark knowledge.** Despite of a notion inspired from dark matter, structured dark knowledge (SDK) is not a Schrodinger's cat. We know that dark knowledge indicates the wrong information about groundtruth yet its ambiguity is potentially helpful for learning. Under our context, it is explained by the discrepancy between $P_w$ and $P(; S)$ across training set, thus, the value of $D_{\text{SDK}}(\boldsymbol{z}_i, w; S)(\forall \boldsymbol{z} \in \mathbb{D})$. We investigate $\frac{D_{\text{SDK}}(\boldsymbol{z}_i, w; S)}{C}$ in average, and see how it changes as the curriculum strategies (PC and SPL) proceed. As shown in Fig.3 , we find that no matter of PC or SPL, they start with hight and vibrated SDK values at the very beginning. It means that, the initial subset selection results are quite different with the teaching decision from CLs. Interestingly, as the training proceeds, the

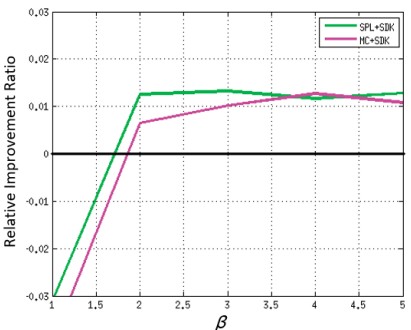

Figure 4: The change of relative performance improvement as we alter the balance parameter $\beta$.

values are progressively become small and steady. On one hand, it is explained by our our progression manner of increasing $k$. On the other hand, it also demonstrates that, the structured knowledge can be "absorbed" by the curriculum-routed teacher underlying SPL and PC.

Besides, since the hyper-parameter $\beta$ controls the knowledge-curriculum balance, we are able to observe SDK via altering $\beta$ in Eq.7 . Our hyper-parameter sensitivity analysis is shown in Fig.4 and for a fair comparison, we specify the evaluation when the learner just consumes $2 \times 10^6$ training instances. The measure (Relative Improvement Ratio) demonstrates the enhanced magnitude compared with the original PC and SPL strategies. Specifically, we find that when $\beta$ is small, CLs equipped with SDK shows negative performance gain, thus, indicates that SDK will be harmful to the learners in this situation. It is because that, when $\beta \to 0$, the selected subset become totally data-driven. Vividly speaking, the "voice" from the curriculum-routed teacher could not be heard by the teacher of teacher. So directly urging the teaching to obey this knowledge might bring about the structured information yet not suitable to the learners. It potentially spoils the training procedure.

Table 2: DCC-based Digit DA across curricula and structured dark knowledge (SDK). (C) and (BP) denote *Concept* and *BP function* as the structured dark knowledge priors.The result in bracket indicates the reproduced performances.

| SDK | | SVHN→MNIST | | | MNIST→USPS | | |
|---|---|---|---|---|---|---|---|
| | | No curriculum | SPL | SPCL | No curriculum | SPL | SPCL |
| Non | | 68.1(69.2) | 67.5 | 70.3 | 79.1(79.0) | 74.4 | 78.5 |
| (C) | source | - | 68.9 | 71.2 | - | 76.9 | 77.9 |
| | target | - | 69.2 | 71.3 | - | **79.9** | 81.8 |
| (BP) | source | - | 70.4 | **73.2** | - | 76.4 | 79.4 |
| | target | - | **71.7** | 73.0 | - | 78.7 | **82.6** |

Table 3: The accuracies (%) of Curriculum-Improved DCC and other DA baselines.

| | Other DA baselines | | DCC+Curricula+SDK (ours) | | | | |
|---|---|---|---|---|---|---|---|
| | RevGred | MMD | No | SPL(C) | SPL(BP) | SPCL(C) | SPCL(BP) |
| **SVHN→MNIST** | 71.1 | 71.1 | 68.1 | 69.2 | 71.7 | 71.3 | **73.2** (+5.1) |
| **MNIST→USPS** | 77.1 | - | 79.1 | 79.9 | 78.7 | 81.8 | **82.6** (+3.5) |

## 4.3 Unsupervised Domain Adaptation.

Provided a source domain with labeled data, domain adaptation (DA) aims to transfer their semantic to classify the unlabeled data from a target domain (non-i.i.d with the source) Long et al. (2015).

**Benchmarks.** Our DA considers two transfer tasks: SVHNNetzer et al. (2011)→MNISTLeCun et al. (1998) and MNIST→USPS. For each transfer, training sets in source and target are used and the evaluation is based on their target test sets.

**DA methods and learner model.** There are plenty of adversarial DA approaches and we choose DDC Tzeng et al. (2017) as our based DA method to present the combination power of curriculum and

Table 4: LSTM net performs sequence2sequence task to calculate decimal numbers. The evaluation is the neccesary epochs to provide the first shot evaluation accuracy more than 99%. The number is the average of five times repeated evaluation. Less means better.

| Methods | Random | Manual | ACL | ACL+SDK(our) |
|---------|--------|--------|-----|--------------|
|         | 24894.6 | 14583.2 | 13357.9 | **13178.6** |

knowledge. To be convenient of curricula and structured knowledge implementation, we reproduce DDC in PyTorch platform. We employ the same architectures of feature extractor, discriminator and classifier, and Adam Kingma & Ba (2014) with the identical hyper-parameter setup has been adopted during the stochastic optimization. We promise its reproduced results as close as possible to their report in the original paper.

**Curricula setup.** SPL used to be a powerful tool in DA Tang et al. (2012). For this consideration, we choose SPL and SPCL as our experimental base teachers to evaluate our T2T. Their curricula act on the DDC's adversarial confusion losses instead of classification losses. It helps to discover whether curricula and knowledge take the positive effects in domain transfer rather than the classifiers.

**Knowledge setup.** Our DA experiment focuses on the structured knowledge including diversity and complementarity. In specific, Concept and *BP function* (Eq.39) are employed as the priors $F(S)$. Eq.7 under these priors are solved by Local Search and GreedMAX (Algorithm.3) respectively.

EMPIRICAL RESULTS AND IMPLICATIONS.

To illustrate more information, the curriculum-routed teachers and their SDK-armed variants are evaluated on the target and source domains, respectively. All the DA results are presented in Table.2, 3 . From Table.2 we discover that, SPL and SPCLs perform even worse than the learners without using curricula. It means that, CL methods (vanilla SPL at least ) might not always take positive effects to guide better training. Such observation is consistent with some previous findings Sachan & Xing (2016); Fan et al. (2018) yet seemingly collides with Tang et al. (2012). One possible explanation is that, DA in Tang et al. (2012) does not involve deep representation learning. So the training loss as difficulty index might be more trustful than those obtained in the modern cases. Thankfully, after injected by structured knowledge among data, SPL and SPCL have been made great again in DA.

Besides, compared with screening source-domain information, SPL and SPCL are better in reweighting examples from the target domain (three in the four best results come from reweighting target data). In SVHN→MNIST, their performances on the target domain are close to the source-based results. In MNIST→USPS experiment, SPL and SPCL perform on the target domain by obvious margins. Under those situations, the complementarity can be a good partner of diversity in DA, which always boosts the learner solely preserving the diversity-based knowledge. Finally, we compare the CL-guided performances with some other famous DA baselines, e.g., RevGred Ganin & Lempitsky (2015), MMD Long et al. (2015), in Table.3. DCC is not born to be the best in these methods, however, by model-agnostic learning with the cooperation of curricula and knowledge, still possible to outperform the other baselines.

### 4.4 SEQUENCE LEARNING.

In this section, T2T assists BCD number calculation Zaremba & Sutskever (2014) and stock price prediction Sun et al. (2014). Different from the previous studies, the structured knowledge considered here comes from the existent facts instead of the oracle knowledge. More are detailed in Appendix.C.

**Learner models.** We employ ordinary Long-Short-Term-Memory (LSTMs) as our student networks. For the calculation experiment, LSTM performs as a sequence-to-sequence model. For the stock price prediction, it receives the price window trajectory to regress the price in the future.

**Curricula setup.** Decimal number calculation can be treated as a case of multi-task learning, where the length of the output decides the task species. For the stock price regression task, we treat the stocks from different industries belonging to different tasks, which also indicates a multi-task setting. So we apply ACL as the only CL teacher algorithm in this two sequence learning cases.

**Knowledge setup.** Though we consider multi-task settings, our structured priors are still based on the relationships among training data. We follow the policy-weight interaction manner in Appendix.A, namely, ACL is used to select tasks and tune their examples' learning weights by Eq.8. As for our knowledge constructions, we apply the hamming distance between the decimal numbers then construct Concept among training instances. Since the stock price data might imply causality, we investigate them by formulating a casual subset selection using the structured prior in Eq.50 .

### DECIMAL NUMBER CALCULATION.

The results of our human-imitated calculation is illustrated in Table.4 . As can be observed, curricula behave quite efficient to speed up the LSTM calculator training. Manual curriculum is very competitive, yet the automatic selection in ACL is even more impressive than the hand-crated difficulty. Finally, SDK is able to help ACL perform the uppermost performance.

### STOCK PRICE PREDICTION.

The empirical studies of stock price prediction are generally based upon two evaluations. In the first evaluation, LSTM are designed to regress the daily Highest and Lowest prices, and we compare the regression error distributions between ACL and ACL+SDK. The results are exhibited in Appendix.C.

Our second evaluation is the backtesting in the real world. Roughly speaking, we specify a buy-and-sell rule and apply it to simulate the trading according to the price predictions from the following three investment strategies: LSTM, LSTM+ACL, LSTM+ACL+SDK, and their portfolio return ratios are weekly accumulated in those years for testing. As illustrated in Fig.5 , the vanilla LSTM always earns the least profits and not recommendable to investors. LSTM+ACL initially attains the highest return ratio, but in the longterm, LSTM+ACL+SDK always outperforms the other investment strategies.

## 5  CONCLUSION AND FUTURE WORK

Teaching to Teach (T2T) by Structured Dark Knowledge (SDK) concerns how to use structured knowledge among training instances to generally enhance existing CL approaches. It connects with a variety of structures represented by generic set functions and refers to diverse model-agnostic curriculum-routed teaching paradigms. Our empirical studies have demonstrated its efficiency across different learning tasks.

There are several branches of development about our research in the future. Firstly, T2T paradigm hitherto focuses on supervised learning, but lack of a glimpse about how to train an agent by exploring structured knowledge among the intelligent agents' experience. How to define a proper structured knowledge and use them to improve curriculum-based reinforcement learning will be quite intriguing. Moreover, though our paradigm attempts to discover sub-structured knowledge, the structured knowledge prior is still predefined. The invention of a complete knowledge discovery algorithm is very promoting. Finally, T2T is an innovative concept about machine education, which should not be limited to structured knowledge. Its development probably leads to more general and efficient teaching paradigms in the high level.

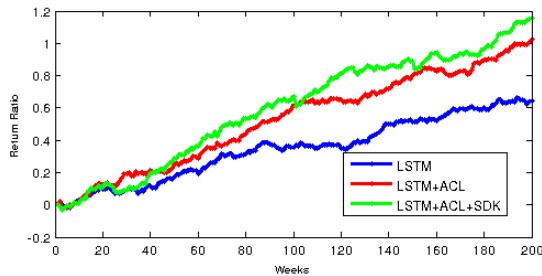

Figure 5: The cumulative return ratios in the market backtest of the following investment strategies: LSTM, LSTM+ACL and LSTM+ACL+SDK.

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

APPENDIX.A

In this Appendix, we revisit the existing major ICL strategies, and consider how to present their explicit weight functions for our methodology.

SELF-PACED LEARNING (SPL) AND VARIANTS

Self-paced learning (SPL) Kumar et al. (2010) accepts training loss $\mathcal{L}(\boldsymbol{z}; \boldsymbol{\theta})$ as the feedback and devise a thread of scheme functions to measure the weight. It typically presents as

$$\min_{\{w_i\}_{i=1}^N, \boldsymbol{\theta}} \sum_{i=1}^{N} w_i \mathcal{L}(\boldsymbol{z}; \boldsymbol{\theta}) + \mathcal{G}(\{w_i\}_{i=1}^N; \lambda) \ \ s.t. \forall i \ \in [N]^+, \ w_i \in [0, 1] \tag{10}$$

where negative function $\mathcal{G}(\{w_i\}_{i=1}^N; \lambda)$ named *self-paced functioin*, controls the pace to incorporate $\mathcal{X}$ by increasing age parameter $\lambda$. Jiang et al. Jiang et al. (2015) provides a definition of $\mathcal{G}(\{w_i\}_{i=1}^N; \lambda)$ in convexity:

**Definition 5.1.** *(Self-paced function ) Controlling the age $\lambda$ in the iterative process, regularizer $\mathcal{G}(\{w_i\}_{i=1}^{|\mathbb{D}|}; \lambda)$ is a self-paced function, which satisfies three principles as follows:*

*(1). $\mathcal{G}(\{w_i\}_{i=1}^{|\mathbb{D}|}; \lambda)$ is convex with respect to $\forall i \in [N], w_i \in [0, 1]$.*
*(2). When all variables are fixed except for $w_i$, $w_i^*$ decreases with $\mathcal{L}(\boldsymbol{z}_i; \theta)$, and holds that $\lim_{\mathcal{L}(\boldsymbol{z}_i; \boldsymbol{\theta}) \to 0} w_i = 1$, $\lim_{\mathcal{L}(\boldsymbol{z}_i; \boldsymbol{\theta}) \to \infty} w_i = 0$.*
*(3). $|\{w_i\}_{i=1}^{|\mathbb{D}|}|$ increases with respect to $\lambda$, holding that $\forall i \in [N], \lim_{\lambda \to 0} w_i^* = 0, \lim_{\lambda \to \infty} w_i^* = 1$.*

where $\{w_i^*\}_{i=1}$ denotes the optimal of latent weights at each step in SPL.

The principles above state how Eq.(10) operates in the iterative training process. Principle 2 indicates that when the age $\lambda$ is certain, SPL inclines to conduct larger latent weight to easy data (with less losses) in favor of complicated ones (with larger losses). Principle 3 promises that as $f$ gets more "mature" by reasonably increasing. $\lambda$, Eq.(10) should accept more complex data into training. These two principles ensure the learning scheme follow in order and advance step by step (*self-paced manner*), and to some extent, the first principle drives the model to pursue a good $\{w_i\}_{i=1}^N$. When $\mathcal{L}(\cdot)$ is convex in $\theta$, Eq.(10) turns into a biconvex optimization, and can be efficiently solved by alternate convex search (ACS).

Obviously, SPL implies the weight function as

$$w(\boldsymbol{z}_i; f_{\boldsymbol{\theta}'}) = \arg \min_{w_i \in [0,1]} \{w_i \mathcal{L}(\boldsymbol{z}; \boldsymbol{\theta}') + \mathcal{G}(\{w_i\}_{i=1}^{|\mathbb{D}|}; \lambda)\} \tag{11}$$

to manage the training distribution. Using different scheme functions $\mathcal{G}(\{w_i\}_{i=1}^N; \lambda)$, we can further specify the weight function for implementation.

- *Binary scheme.* Given $\mathcal{G}(\{w_i\}_{i=1}^N; \lambda) = -\lambda \sum_{i=1}^N w_i$,

$$w(\boldsymbol{z}; f_{\boldsymbol{\theta}'}) := \begin{cases} 0, & \mathcal{L}(\boldsymbol{z}; \boldsymbol{\theta}') > \lambda \\ 1, & \mathcal{L}(\boldsymbol{z}; \boldsymbol{\theta}') \leq \lambda \end{cases} \tag{12}$$

- *Linear scheme.* Given $\mathcal{G}(\{w_i\}_{i=1}^N; \lambda) = \frac{1}{2}\lambda \sum_{i=1}^N w_i^2 - 2w_i$,

$$w(\boldsymbol{z}; f_{\boldsymbol{\theta}'}) := \begin{cases} 0, & \mathcal{L}(\boldsymbol{z}; \boldsymbol{\theta}') > \lambda \\ 1 - \dfrac{\mathcal{L}(\boldsymbol{z}; \boldsymbol{\theta}')}{\lambda}, & \mathcal{L}(\boldsymbol{z}; \boldsymbol{\theta}') \leq \lambda \end{cases} \tag{13}$$

- *Logarithmic scheme.* Given $\mathcal{G}(\{w_i\}_{i=1}^N; \lambda) = \sum_{i=1}^N (1 - \lambda)w_i - \frac{(1-\lambda)^{w_i}}{\log(1-\lambda)}$, $\lambda \in (0, 1)$

$$w(\boldsymbol{z}; f_{\boldsymbol{\theta}'}) := \begin{cases} x = 0, & \mathcal{L}(\boldsymbol{z}; \boldsymbol{\theta}') > \lambda \\ y = \dfrac{\log(\mathcal{L}(\boldsymbol{z}; \boldsymbol{\theta}') + 1 - \lambda)}{1 - \lambda}, & \mathcal{L}(\boldsymbol{z}; \boldsymbol{\theta}') \leq \lambda \end{cases} \tag{14}$$

- *Mixture scheme.* Given $\mathcal{G}(\{w_i\}_{i=1}^N; \lambda) = \frac{\lambda\rho}{1-\rho} \sum_{i=1}^N \log(w_i + \frac{\rho}{1-\rho})$, $\rho \in (0, 1)$,

$$w(\boldsymbol{z}; f_{\boldsymbol{\theta}'}) := \begin{cases} 0, & \mathcal{L}(\boldsymbol{z}; \boldsymbol{\theta}') > \lambda \\ 1, & \mathcal{L}(\boldsymbol{z}; \boldsymbol{\theta}') \leq \rho\lambda \\ \dfrac{\rho\lambda(\lambda - \mathcal{L}(\boldsymbol{z}; \boldsymbol{\theta}'))}{(1 - \rho)\lambda\mathcal{L}(\boldsymbol{z}; \boldsymbol{\theta}')}, & \rho\lambda < \mathcal{L}(\boldsymbol{z}; \boldsymbol{\theta}') \leq \lambda \end{cases} \tag{15}$$

ICLs with Policy Sampling

Policy sampling CL Graves et al. (2017) Matiisen et al. (2017) Doan et al. (2018) employ a multi-armed bandit (MAB) as the sampling tool. In their settings, each group of samples or task has been treated as a slot, which is decided the sample to chose or not in each step mini-batch sampling. Although these branch of method consider task as selection instance. In our scenario, it can be viewed as a sample and task pair to merge in our framework, thus, we define $w(z; f_\theta)$ as the task-specific weight function. Training instances in the same task are assigned the same weight value.

**Automated curriculum learning (ACL).** Graves et al. (2017) first leverages adversarial MAB algorithm to solve the predefined curricula issues. Specifically, they treat each slot in the bandit as a task[4], then apply EXPS.3 algorithm to select them to ensemble mini-batches. Concretely, suppose we have $M$ tasks for consideration and $\forall i \in [M]$, the $\mathbb{D}_i$ corresponds to its subset ($\bigcup_{i=1}^{M} \mathbb{D}_i = \mathbb{D}$). The strategy in ACL to select an instance $z$ is

$$\pi_{i,t}^{\text{EXP3.S}} := (1 - \mu) \frac{e^{\mathbf{w}_t^{(i)}(\theta)}}{\sum_{j=1}^{M} e^{\mathbf{w}_t^{(j)}(\theta)}} + \frac{\mu}{M}, \quad \forall z \in \mathbb{D}_i \tag{16}$$

where $\mu \in (0, 1)$ is the exploration ratio and $\mathbf{w}_t^{(i)}(\theta)$ is the accumulated reward that.

$$\mathbf{w}_{i,t}(\theta') := \log \left[ t^{-1} \exp\{\mathbf{w}_{i,t-1} + r_{i,t-1}(z, \theta')\} + (1 - t^{-1}) \frac{\sum_{j \neq i} \exp\{\mathbf{w}_{j,t-1} + r_{j,t-1}(z, \theta')\}}{N - 1} \right] \tag{17}$$

where $\forall i \in [M]$, $\mathbf{w}_{i,0} = 0$. The $t^{th}$ iteration reward $r_{i,t-1}(z, \theta')$ in ACL comes from diverse learning progress signals $v$. They are calculated online, trimmed into the range $[-1, 1]$ by the following rule:

$$r_{i,t}(z, \theta') := \begin{cases} -1, & \frac{v}{\tau(z)} < q_t^{\text{lo}} \\ 1, & \frac{v}{\tau(z)} > q_t^{\text{hi}} \\ \frac{\frac{v}{\tau_t(z)} - q_t^{\text{lo}}}{q_t^{\text{hi}} - q_t^{\text{lo}}}, & q_t^{\text{lo}} \leq \frac{v}{\tau(z)} \leq q_t^{\text{hi}} \end{cases} \tag{18}$$

where $\tau_t(z)$ denotes the longest time to process $\forall z \in \mathbb{D}$ in the iteration $t$; $q_t^{\text{lo}}$ and $q_t^{\text{hi}}$ denote the trimmed reward lower and upper bounds in the $t^{th}$ iteration (the authors set $20\%$ and $80\%$ quantile of $\{\frac{v_i}{\tau(z_i)}\}_{i=1}^{|\mathbb{D}|}$). The learning progress signals are mainly classified into two branches: *loss-driven* and *complexity-driven*. The former focuses on the loss reduction on each instance; the latter consider the model complexity variation. In this paper, we focus on the loss-driven progress signals empirically showing more superior than complexity-driven progress signals in Graves et al. (2017).

- *Prediction Gain* (PG). PG aims to estimate the change of loss of instance $z$. The more decrease indicates $z$ is easier to learn in this step:

$$v_{\text{PG}} = \mathcal{L}(z; \theta') - \mathcal{L}(z; \theta) \tag{19}$$

- *Gradient Prediction Gain* (GPG). PG requires two forward propagation to obtain the current loss and previous loss of the considered instances. Developed from the first-order Taylor series approximation to PG, GPG only needs one forward operation:

$$v_{\text{GPG}} = ||\nabla_\theta \mathcal{L}(z; \theta)||_2 \tag{20}$$

- *Self-Prediction Gain* (SPG). PG and GPG are the biased estimate of the change of loss when the parameter change from $\theta$ to $\theta'$ is caused by $z$. To mitigate this issue, SPG select the other sample $z'$ from the same task to replace $z$:

$$v_{\text{SPG}} = \mathcal{L}(z'; \theta') - \mathcal{L}(z'; \theta), \quad z' \sim \mathbb{D}_{i(z)} \tag{21}$$

where $\mathbb{D}_{i(z)}$ indicates task-specific or category-specific training instance subset.

---

[4]The original paper consider two cases: multi-task learning and single-task learning by previously learning the other tasks as the bridge. Here we only consider the first case.

To bridge the weight $w_i$ to $\pi_i$, it needs to clarify the structured knowledge is built on the relations among instances or taskes.

If it is built on the relations among instances, since ACL performs on multiple tasks or categories, where each instance is uniformly sampled. So we need to do just set $w_i = 1 (\forall z_i \in \mathbb{D})$ in $P_w$ and update the balance strategy through Eq.8. To perform the refined curriculum, we first apply $\pi_{i,t}^{\text{EXP3.S}}$ to choose a task slot, then uniformly sample an instance and assign its weight by the balance strategy close-form result.

If it is built on the relations among tasks or categories, then each instance corresponds to a task or category. In this case, $w_i$ might be directly treated as $\pi_{i,t}^{\text{EXP3.S}}$.

In our evaluation of ACL, we only consider the first case.

**Teacher-student curriculum learning (TSCL)**. TSCL can be viewed as a simple version of ACL:

$$\log \pi_i(z; f_\theta) = \alpha r_i(z, f_{\theta'}) + (1 - \alpha) \log \pi_i(z; f_{\theta'})$$

where $\pi_i(z; f_{\theta'})$ is the bandit sampling strategic probability; $r(z, f_{\theta'})$ is the sum of changes in the evaluation scores of task-based instance $z$. $\alpha$ denotes the discounted ratio to calculate the moving average of $r(z, f_{\theta'})$. Obviously, the teacher-student strategy focus on task selection and training instances in the same task are uniformly collected.

$w(z; f_\theta)$ connects $\pi_i(z; f_\theta)$ under the same rule of ACL.

LEARNABLE ICLs.

Recent triumphs in advanced machine learning raise a question to machine teaching methods: Is it possible to devise a guiding curriculum via a neural network? MentorNet Jiang et al. (2017), ScreenNet Kim & Choi (2018) and L2T Fan et al. (2018) agree on this total automatic manner. They apply different networks (MentorNet use RNN, Screener Net proposed a CNN-base iterative paradigm, L2T use deep reinforcement learning agent) to achieve this goal. In our paper, we deliver simple introductions of MentorNet and sampling-based L2T.

**MentorNet.** MentorNet is motivated by SPL family. SPL family predefines a self-paced function and basically receive training losses as difficulties. To mitigate the arbitrary difficulty design, MentorNet performs as $w_\phi(g_m(z_i)) \in [0, 1]$ ($f_m(z_i)$ denotes the performance feature of $z_i$. It is constructed by multiple validation index. More refers to Jiang et al. (2017)), which produces a value to substitute inferred weight $w_i$ in SPL. The objective can be generally formulated as:

$$\min_{\phi,\theta} \sum_{z_i \in \mathbb{D}} w_\phi(g_m(z_i)) \mathcal{L}(z_i; \theta) + \mathcal{G}(\{w_\phi(g_m(z_i))\}_{i=1}^{|\mathbb{D}|}; \lambda) \tag{22}$$

Obviously, this objective is developed from Eq.11. But $\mathcal{G}$ might refer to non-convex property. After imported structured dark knowledge $D_{SDK}$, the objective is refined as:

$$\min_{\phi,\theta} \sum_{z_i \in \mathbb{D}} \left[ w_\phi(g_m(z_i)) \mathcal{L}(z_i; \theta) \right.$$
$$\left. + \gamma D_{\text{SDK}}(z_i, w_\phi; S) \right] + \mathcal{G}(\{w_\phi(g_m(z_i))\}_{i=1}^{|\mathbb{D}|}; \lambda) \tag{23}$$

then $S$ keeps alternatively updating as our Algorithm performs.

**Sampling-based L2T.** Different from MentorNet, L2T directly employ a policy network to select training instance. It first specifies a state feature $g_s(z)$ of instance $z$, then formulate the curriculum-based teaching as a binary classification decision on each instance. These decisions are made by a policy network ($w_\phi$ in our context) trained through a policy gradient algorithm REINFORCE Williams (1992).

Concretely, suppose the final activation of $w_\phi$ is a sigmoid function, thus, $Policy_\phi(a|g_s(z)) = a w_\phi(g_s(z)) + (1 - a)(1 - w_\phi(g_s(z)))$, where $a \in \{0, 1\}$ indicates the selection decision. Then the policy gradient update performs as

$$\sum_{z \in \Xi} \nabla_\phi \log[a w_\phi(g_s(z)) + (1 - a)(1 - w_\phi(g_s(z)))] r(a, g_s(z)) \tag{24}$$

where $\Xi$ denotes the training episode, $r(a, g_s(\boldsymbol{z}))$ denotes the sampled estimation of the teaching reward from one episode execution (More details refer to Fan et al. (2018)). For structured dark knowledge transfer, the update gradient has turned into

$$\sum_{\boldsymbol{z} \in \Xi} \nabla_\phi \big[ \log[a w_\phi(g_s(\boldsymbol{z})) + (1-a)(1 - w_\phi(g_s(\boldsymbol{z})))] r(a, g_s(\boldsymbol{z})) - \gamma D_{\text{SDK}}(\boldsymbol{z}, w_\phi; S) \big] \quad (25)$$

## APPENDIX.B

In this Appendix, we specify our structured knowledge prior $F$, namely, a set function over $\mathbb{D}$.

Assume the training set $\mathbb{D}$ is the ground set we consider. Set function $F : 2^{\mathbb{D}} \to \mathbb{R}$ can be viewed as a combinatorial optimization objective, which searches a subset $S \subseteq \mathbb{D}$ to maximize/minimize the value of $F(S)$. In specific in Eq.7, we consider the subset selection with cardinality constraint, where $|S| \le k$. Moreover, we only consider the maximization, due to minimization can be achieved by setting $-F(S)$.

### SUBMODULARITY AND SUPERMODULARITY.

General subset selection is a NP-hard problem, yet when $F$ satisfies *submodularity* Fujishige (2005), there are many effective algorithms providing the sub-optimal results with $\epsilon$-approximation Nemhauser et al. (1978) Zhou & Spanos (2016b). Namely, suppose $S^*$ denotes the global optima, the $\epsilon$-approximate sub-optima $\hat{S}$ maintains $S^* \le \epsilon \hat{S}$.

**Definition 5.2 (submodularity).** *Set function $F : 2^{\mathcal{D}} \to \mathbb{R}_+$ is submodular set function if*

$$F(S_1) + F(S_2) \ge F(S_1 \cup S_2) + F(S_1 \cap S_2), \quad s.t. \ \forall \ S_1, \ S_2 \subset \mathcal{D} \quad (26)$$

**Definition 5.3 (submodularity (*deminishing return*)).**

$$F(S_1 \cup \{\boldsymbol{z}\}) - F(S_1) \ge F(S_2 \cup \{\boldsymbol{z}\}) - F(S_2), \quad s.t. \ \forall \ S_1 \subset \ S_2 \subset \mathcal{D}, \boldsymbol{z} \in \mathcal{D}/S_2 \quad (27)$$

The definitions above are equivalent. Besides, if $F(\cdot)$ is submodular, then $-F(\cdot)$ is *supermodular*. $F$ can be submodular and supermodular at the meantime, then we get $F(S_1) + F(S_2) = F(S_1 \cup S_2) + F(S_1 \cap S_2)$ and $F(\cdot)$ is termed *modular function*. $F(\cdot)$ is *normalized* if $F(\emptyset) = 0$; is *monotonically increases* if $F(S_1) < F(S_2)$ iff $\forall S_1 \subset S_2$; *monotonically decreases* if $F(S_1) < F(S_2)$ iff $\forall S_2 \subset S_1$.

**Remark 5.1.** *If $F(S)$ is submodular/supermodular, then Eq.7 is submodular/supermodular maximization under the cardinality constraints.*

Since the second term of Eq.7 is modular, the remark is obvious.

**Remark 5.2.** *If $F(S)$ is supermodular, Eq.7 turns to a submodular minimization problem under cardinality constraint.*

**Remark 5.3.** *If $\forall j \in [J]$, $F_j(S)$ is a submodular/supermodular function, then*

$$F(S) = \sum_{j=1}^{J} \alpha_j F_j(S), \ \forall j \in [J], \ \alpha_j > 0 \quad (28)$$

*is also a submodular/supermodular function.*

**Diversity and Complementarity.** Submodularity and supermodularity convey different structured properties. Under the context of maximization, submodular function enforces the subset selection presenting diversity and summarization. For example, if there is a document to be summarized by $k$ sentences, we can maximize a submodular function about its organized structure to achieve the goal. On the other hand, supermodularity indicates cooperation and similarity, namely, the group of data appear more coherent are more probably selected together.

If $F(S)$ is submodular, then $\max\limits_{S \subseteq \mathbb{D}, |S| \le k} \mathcal{J}(S)$ can be solved by Local Search (Algorithm.3). The algorithms are guaranteed to approximate the global optima.

**Remark 5.4.** *Local Search provides a sub-optimal result $\hat{S}$ satisfying $\max\limits_{S \subseteq \mathbb{D}, |S| \le k} \mathcal{J}(S)$ by $\mathcal{J}(S^*) \le$ $(\frac{1 - e^{-\kappa_{\mathcal{J}}}}{\kappa_{\mathcal{J}}}) \mathcal{J}(\hat{S})$.*

where $\kappa_{\mathcal{J}}$ denotes the curvature of $\mathcal{J}$:

**Definition 5.4** (**submodular curvature** Fujishige (2005)). *$\kappa_F$ is the curvature of submodular function F:*

$$\kappa_F = 1 - \min_{\boldsymbol{z} \in S \subseteq \mathbb{D}} \frac{F(S) - F(S/\{\boldsymbol{z}\})}{F(\{\boldsymbol{z}\})} \tag{29}$$

---

| **Algorithm 2** Local Search Nemhauser et al. (1978) | **Algorithm 3** GREEDMAX Bai & Bilmes (2018) |
|---|---|
| $A_0 \leftarrow \emptyset, \mathbb{D}, k, \mathcal{J}(\cdot)$. | $A_0 \leftarrow \emptyset, B_0 \leftarrow \mathbb{D}, k, F_1(\cdot), F_2(\cdot)$. |
| **for** $t = 1$ to $k$ **do** | **for** $t = 1$ to $k$ **do** |
| $\quad \boldsymbol{z}^{(t)} \leftarrow \arg \max_{\boldsymbol{z} \in \mathbb{D}/A_t} \left[ \mathcal{J}(A_t \cup \{\boldsymbol{z}\}) - \mathcal{J}(A_t) \right]$. | $\quad \boldsymbol{z}^* = \arg\max_{\boldsymbol{z} \in B_t} h(\boldsymbol{z}) + F_1(A_{t-1} \cup \{\boldsymbol{z}\}) - F_1(A_{t-1}) + F_2(A_{t-1} \cup \{\boldsymbol{z}\}) - F_2(A_{t-1})$ |
| $\quad A_t \leftarrow A_{t-1}$ | $\quad A_t \leftarrow A_{t-1} \cup \{\boldsymbol{z}^*\}, B_t \leftarrow B_{t-1}/\{\boldsymbol{z}^*\}$ |
| **end for** | **end for** |
| **return** $A_k$. | **return** $A_k$. |

---

If $F(S)$ is supermodular, $\max_{S \subseteq \mathbb{D}, |S| \leq k} \mathcal{J}(S)$ is reduced to size-constrained submodular minimization. This problem can be approximately solved by a sampling-based algorithm Svitkina & Fleischer (2011) while the constant factor polynomial-time approximation algorithm is non-exist in this problem. But in our setting, we pay more interest in the constrained suBmodular+suPermodular (BP) maximization Bai & Bilmes (2018) rather than a pure supermodular maximization. BP maximization is more realistic and can be easily extended from a pure supermodular maximization.

**Definition 5.5** (**BP maximization**). *Given a normalized monotonically increasing submodular function $F_1$ and supermodular function $F_2$, BP maximization refers to solve a following objective*

$$\max_{S \subseteq \mathbb{D}, |S| \leq k} F(S) = F_1(S) + F_2(S) \tag{30}$$

The meaning of BP maximization is developed from diversity. In structured knowledge with diversity, we hope to choose data distributed along with the structure, thus, as sparse as possible. But in some situations, we also hope some certain groups of data are selected together. The cooperation of $F_1$ and $F_2$ is able to produce this effect.

**Remark 5.5.** *Provided $F$ as Definition.5.5, $\max_{S \subseteq \mathbb{D}, |S| \leq k} \mathcal{J}(S)$ is a BP maximization.*

*Proof.* We know the second term of $\mathcal{J}(S)$ is modular and monotonically increases. then

$$\mathcal{J}(S) = F_1(S) + F_2(S) + \beta \sum_{\boldsymbol{z}_i \in S} h(\boldsymbol{z}_i)$$

$$= \underbrace{F_1(S) + \beta \sum_{\boldsymbol{z}_i \in S} h(\boldsymbol{z}_i)}_{\substack{\text{normalized monotonically increasing} \\ \text{submodular function}}} + F_2(S)$$

where $h(\boldsymbol{z}_i) = 1 - \frac{D_{\text{SDK}}(\boldsymbol{z}_i, w; S)}{C}$. Proof is concluded. $\qquad \square$

**Remark 5.6.** *BP maximization solved by GreedMAX (Algorithm.3) leads to an approximate outcome as.*

$$(1 - \kappa^{F_2})\left(1 + \frac{(\kappa_{F_1})^2(1 - \kappa^{F_2})^2}{6(\beta c + 1)^2}\right)\mathcal{J}(\hat{S}) \geq \mathcal{J}(S^*), \ s.t. \ c = \min_{\boldsymbol{z}_i \in S} \frac{h(\boldsymbol{z}_i)}{F_1(\{\boldsymbol{z}_i\})} \tag{31}$$

where $\kappa_{F_1}$ and $\kappa^{F_2}$ respectively indicate the curvatures of submodular function $F_1$ and supermodular function $F_2$:

**Definition 5.6** (**supermodular curvature**). *Given a monotonically increasing supermodular function $F_2$, the curvature of $F_2$ $\kappa^{F_2}$ is defined as*

$$\kappa^{F_2} = 1 - \min_{\boldsymbol{z} \in S} \frac{F_2(\{\boldsymbol{z}\})}{F_2(S) - F_2(S/\{\boldsymbol{z}\})} \tag{32}$$

*Proof.* Remark.5.6 comes from the theoretical result of Bai & Bilmes (2018):

**Lemma 5.1.** *Given the maximization problem in Eq.30, GreedMAX obtain a suboptimal $\hat{S}$ that maintains an approximation toward $S^*$:*

$$\frac{1}{\kappa_{F_1}}(1 - e^{-(1-\kappa^{F_2})\kappa_{F_1}})\big(F_1(\hat{S}) + F_2(\hat{S})\big) \geq \big(F_1(S^*) + F_2(S^*)\big) \tag{33}$$

then by Remark.5.5, we have

$$\frac{1}{\kappa_{F_1+\beta\sum h_i}}(1 - e^{-(1-\kappa^{F_2})(\kappa_{F_1+\beta\sum h_i})})\mathcal{J}(\hat{S}) \geq \mathcal{J}(S^*) \tag{34}$$

where $\kappa_{F_1+\beta\sum h_i}$ denotes the submodular curvature of $F_1(S) + \beta \sum_{z_i \in S} h(z_i)$. Then

$$
\begin{aligned}
\kappa_{F_1+\beta\sum h_i} &= 1 - \min_{z_i \in S} \frac{\beta h(z_i) + F_1(S) - F_1(S/\{z_i\})}{\beta h(z_i) + F_1(\{z_i\})} \\
&= \max_{z_i \in S} \frac{F_1(\{z_i\}) - [F_1(S) - F_1(S/\{z_i\})]}{\beta h(z_i) + F_1(\{z_i\})} \\
&= \max_{z_i \in S} \frac{1 - [\frac{F_1(S) - F_1(S/\{z_i\})}{F_1(\{z_i\})}]}{\beta \frac{h(z_i)}{F_1(\{z_i\})} + 1} \\
&\leq \frac{\kappa_{F_1}}{\beta c + 1}, \ c = \min_{z_i \in S} \frac{h(z_i)}{F_1(\{z_i\})}
\end{aligned}
\tag{35}
$$

Besides, we observe the inequality that $\forall x \in [0, 1]$,

$$
\begin{aligned}
1 + \frac{x^2}{6} &\geq 1 - \frac{x}{2} + \frac{x^2}{6} = \frac{1 - (1 - x + \frac{x^2}{2} - \frac{x^3}{6})}{x} \\
&\geq \frac{1 - e^{-x}}{x}
\end{aligned}
\tag{36}
$$

Combine Eq.34, 35 and we have

$$
\begin{aligned}
&(1 - \kappa^{F_2})\big(1 + \frac{(\kappa_{F_1})^2(1 - \kappa^{F_2})^2}{6(\beta c + 1)^2}\big)\mathcal{J}(\hat{S}) \\
\geq &(1 - \kappa^{F_2})\big(1 + \frac{(\kappa_{F_1+\beta\sum h_i})^2(1 - \kappa^{F_2})^2}{6}\big)\mathcal{J}(\hat{S}) \\
\geq &\frac{1 - \kappa^{F_2}}{(\kappa_{F_1+\beta\sum h_i})(1 - \kappa^{F_2})}(1 - e^{-(1-\kappa^{F_2})(\kappa_{F_1+\beta\sum h_i})})\mathcal{J}(\hat{S}) \geq \mathcal{J}(S^*)
\end{aligned}
\tag{37}
$$

. Conclude the proof. $\square$

Here we specify some kind of $F(S)$. They are applied in our first and second experiments to show structured knowledge prior about diversity and complementarity among data.

- *Concepts.* Suppose $\mathbb{D}$ contains $J$ concepts to describe training instances. Each concept can be modeled by a subset $V_j \subseteq \mathbb{D}(\forall j \in [J])$. Then we can formulate this structured knowledge via a positive linear combination of concave over modular functions.

$$F(S) = \sum_{j=1}^{J} \alpha_j |S \cap V_j|^{\mu}, \ \forall j \in [J], \tag{38}$$

$$V_j \subseteq \mathbb{D}, \ \bigcup_{j=1}^{J} V_j = \mathbb{D}, \ \alpha_j > 0, \ 0 < \mu < 1$$

  The implication of this structured knowledge is proposed to evenly absorb the $J$ concepts. The selection criterion of $V_j$ is based on the experiment setting.

- *BP function.* Different from the above submodular set functions, BP function is neither submodular nor supermodular, which indicates the balance between diversity and complementarity. We follow the example in Bai & Bilmes (2018) and present this structured knowledge by

$$F(S) = \sum_{j=1}^{J} |S \cap V_j|^{\mu} + \sum_{j=1}^{I} \max\{0, \frac{|S \cap W_i| - m}{1 - m}\}, m \in (0, 1) \tag{39}$$

where the first and second terms indicate diversity and complementarity respectively. Specifically, $V_j$ denotes a set of instance containing certain semantic meaning, $W_i$ denotes a set of ambiguous instances nearby the optimal decision boundary.

SUBMODULAR INDEX.

Although submodular functions are widely applied in structured inference, there are always some structures that can not be modeled by this property. Hence we are going to discuss a more general conceptual standard named *Submodular Index* (SI). SI provides a view to describe a set function how close to be submodular. It leads to an unified approximate method to solve generic subset selection.

The following theoretical results totally come from Zhou & Spanos (2016a).

**Definition 5.7** (Submodular Index (SI) ). *For a set function $F : 2^{\mathbb{D}} \to \mathbb{R}$ the submodularity index (SI) for a location set $L$ and a cardinality $k$, denoted by $\Omega_f(L, k)$, is defined as*

$$\Omega_f(L, k) := \min_{A \subseteq L, \ S \cup A = \emptyset, \ |S| \leq k} \varphi_f(S, A) \tag{40}$$

where $\varphi_f(S, A)$ indicates local submodular index (LSI):

**Definition 5.8** (Local submodular Index (LSI)).

$$\varphi_f(S, A) := \sum_{\boldsymbol{x} \in S} F(\{\boldsymbol{x}\}|A) - F(S|A) \tag{41}$$

where $F(\{\boldsymbol{x}\}|A) = F(\{\boldsymbol{x}\} \cup A) - F(A)$ and $F(S|A) = F(S \cup A) - F(A)$. Based on this definition, Algorithm.4 is provided to solve generic subset selection.

**Theorem 5.1.** *For a general (possibly non-monotonic, non-submodular) set function $F$, let the optimal solution of the cardinality-constrained maximization be denoted as $S^*$, and the solution $\hat{S}$ of random greedy algorithm satisfying*

$$\mathbb{E}[F(\hat{S})] \geq \left(\frac{1}{e} - \frac{\xi_{\hat{S},k}^f}{\mathbb{E}[F(\hat{S})]}\right) F(S^*) \tag{42}$$

where $\xi_{\hat{S},k}^f = \Omega_{\hat{S},k}^f + \frac{k(k-1)}{2} \max\{0, \Omega_{\hat{S},2}^f\}$

There are two extensions from the theorem above:

**Corollary 5.1.** *For monotonic set functions in general, random greedy algorithm achieves*

$$\mathbb{E}[F(\hat{S})] \geq \left(1 - \frac{1}{e} + \frac{\Omega_f'(\hat{S}, k)}{\mathbb{E}[F(\hat{S})]}\right) F(S^*) \tag{43}$$

*and deterministic version of Algorithm.4 maintains*

---

**Algorithm 4** Random Greedy Zhou & Spanos (2016a)

---

$A_0 \leftarrow \emptyset, \mathbb{D}, k, \mathcal{J}(\cdot).$
**for** $t = 1$ to $k$ **do**
$\quad \boldsymbol{z}^{(t)} \leftarrow \arg \max_{M_t \subset \mathbb{D}/A_{t-1}, |M_t|=k} \sum_{u \in M_t} F(u|A_t).$
$\quad$ Draw $u$ uniformly from $M_t$.
$\quad A_t \leftarrow A_{t-1} \cup \{u\}$
**end for**
**return** $A_k$.

---

$$F(\hat{S}) \geq \left(1 - \frac{1}{e} + \frac{\Omega_f'(\hat{S}, k)}{F(\hat{S})}\right) F(S^*) \tag{44}$$

*where*

$$\Omega_f'(\hat{S}, k) := \begin{cases} \Omega_f'(\hat{S}, k), \ \Omega_f'(\hat{S}, k) > 0 \\ (1 - \frac{1}{e})^2 \Omega_f'(\hat{S}, k), \ \Omega_f'(\hat{S}, k) \leq 0 \end{cases} \tag{45}$$

**Corollary 5.2.** *For submodular function that are not necessarily monotonic, random greedy algorithm has performance*

$$\mathbb{E}[F(\hat{S})] \geq \left(\frac{1}{e} - \frac{\Omega_f(\hat{S}, k)}{\mathbb{E}[F(\hat{S})]}\right) F(S^*) \tag{46}$$

Here we specify some cases of structured knowledge priors $F(S)$ that can be solved by Algorithm.4.

- *Asymmeric Graph Cut.* Unordered graphs are common structures to represent data knowledge. Suppose vertices denote samples in $\mathbb{D}$ and edges reflect the pairwise relations among them, $F(S)$ presents a subconnection based on a RBF kernel variant $\mathbf{K}$ ($\det(\mathbf{K} > 0)$), i.e., $\forall \boldsymbol{z}_i, \boldsymbol{z}_j \in \mathcal{D}, \mathbf{K}_{i,j} = \exp(-\frac{||f_{\theta^*}(\boldsymbol{z}_i) - f_{\theta^*}(\boldsymbol{z}_j)||_2^2}{\sigma^2})$.

$$F(S) = -\mathbf{1}_S^T \mathbf{K} \mathbf{1}_S + \beta \mathbf{1}^T \mathbf{K} \mathbf{1}_S \tag{47}$$

where $\sigma$ is the bandwidth that we set $0.4$ in our experiment. $f_{\theta^*}(\boldsymbol{z})$ indicates a oracle knowledge embedding of $\boldsymbol{z}$. Note that, Eq.47 is not monotonic, thus, it can not be solved by Local Search. In the implementation, it would be more favorable to choose the sparse nearest neighbors and normalize them to construct $\mathbf{K}$.

- *Granger Causality* (**GC**). Consider the random processes $X^{(N)}$ and $Y^{(N)}$, where $X^{(i)} = \{X_1, X_2, \cdots, X_i\}$ ($1 \leq i \leq N$ indicates time index). Directed information is a term defined by

$$\mathcal{I}(X^{(n)} \to Y^{(n)}) = \sum_{t=1}^{n} I(X^{(t)}, Y_t | Y_{t-1}) \tag{48}$$

The formula can be viewed as the aggregated dependence between the history of $X$ and the current value of process $Y$, given the past observations of $Y$. Then in a specific iteration $t$, it conveys a causal relation that given the past $Y_{t-1}$, $X_t$ on $Y_t$ should be unique. A subset selection problem based on Eq.48 is so-called causal subset selection. There are two following specifications about this problems:

**Definition 5.9** (Causal Sensor Placement).

$$F(S) = \mathcal{I}(S^{(n)} \to \overline{S}^{(n)}) \tag{49}$$

**Definition 5.10** (Casual Covariates Selection).

$$F(S; Y) = \mathcal{I}(S^{(n)} \to Y^{(n)}) \tag{50}$$

. The first knowledge prior (Eq.49) is unsupervised, yet the second knowledge prior (Eq.50) presents a target process $Y^{(N)}$ that the selected subset should consider. More information about them can be found in Zhou & Spanos (2016a).

APPENDIX.C

MORE SPECIFIC IMPLEMENTATION OF CLS.

DETAILS IN PC.

For training a learner, PC performs as a stochastic sampling process under the distribution constantly changing in a fixed step. Initially it focuses on easiest examples, then the weights of more difficult examples gradually increase so that finally the distribution become converge to the training prior $P(\boldsymbol{z})$. In our implementation, we schedule the PC decreasing process according to the minimum epochs for the leaner's convergence. Then it linearly changes the proportion of the sampling distributions.

If $w_i$ is updated, PC presents either a reweighting scheme or a renewed sampling distribution. Both of them are available in our methodology. In order to reveal the data mining efficiency compared with the other CL baselines, PC performs in a stochastic sampling style in our experiment. Namely, the mini-batches are sampled by $Q_\lambda(\boldsymbol{z})$ ($\lambda$ indicates the proceeding schedule).

To implement our method on PC, we use $w$ to reweight their instance sampling distribution, and employ the updated training distribution to sample training instances.

DETAILS IN L2T.

Our L2T is reimplemented under the tensorflow platform. We follow the experimental details in the original paper. Then for the hyper-parameter setting, state feature is concatnated by the data, model and combined features, and the evaluation rate threshold $\tau$ is set $0.84$.

Two evaluation settings are demonstrated in L2T. Accordingly, their teaching schedules are quite different from the other CL approaches. The authors claimed that the teacher in L2T co-evolves with the student, yet in fact, teacher requires a pre-training on a student to perform better education quality. So we employ different learning manners to incorporate L2T into our T2T. In the first setting when the teacher is trained on CIFAR-10, our SDK regulates its policy learning, then when the teacher is fixed to teach the student, SDK is deactivated. In the second setting when the teacher is trained on MNIST, our SDK do not performe at this pre-training. Then when the teacher performs teaching on CIFAR-10, SDK performs to finetune the teaching strategy.

TINY IMAGE CLASSIFICATION.

**Benchmark and learner.** The first experiment is conducted on CIFAR-10. CIFAR-10 is widely-used benchmark for visual classification. It contains $60,000$ RGB images with size $32$ across $10$ classes. The data has been partitioned into a training set with $50,000$ images and test set with $10,000$ images. During our training, data augmentation is applied: each images has been padded with 4 pixels to each side and get cropped into $32 \times 32$. We apply ResNet32 models in tensorflow implementation and the size of mini batch is set $128$. We follow the optimization strategy in He et al. (2016). Namely, we employ Momentum-SGD as the solver and set the initial learning rate as $0.1$, which is multiplied by $0.1$ after the $32,000^{th}$ and $48,000^{th}$ update. The training leads to $93.26\%$ in a test accuracy.

**Curricula specification.** We evaluate PC, SPL and L2T in CIFAR10. ACL is not born for single task learning. SPCL is a reweighting approach, yet in this classification experiment, we expect to compare the data selection efficiency. Hence we choose the binary SPL as the representative in SPL family. Their implementations have been previously detailed.

**Knowledge engineering.** The structured knowledge prior is built upon Eq.38 where each index $j \in [J]$ denotes a concept among data. More specifically, $\forall j \in [M]$ we set $\alpha_j = 1$, and we determine $V_j$ by a simple feature selection principle: we collect all training data and perform feature-level $J$-means clustering (the feature lie on oracle knowledge space, namely, extracted from a pre-trained ResNet110). Then each example will be assigned to some of those $J$ clusters that the feature entries of this example belong to. We set $J = 64$ in our experiment and employ the technique in Arthur & Vassilvitskii (2007) to initiate $J$ seeds before clustering.

DOMAIN ADAPTATION.

**Curricula specification.** To ensure the robustness in adversarial learning, SPL and SPCL applied in our DA obey three extra learning principles. First, they only perform the weight inferences during the feature extractors update. Second, at each time when the discriminator update finishes, $K$ is reset, thus, rearranges the mini-batch self-paced process narrated in SubSection.4.1.1 ($E$ is set as maximum the iterations of each feature extractor updating process). Our strategy separately operates on the training sets from source and target domains. Structured dark knowledge is also based on this separation. Different from the classification experiment, the selected training instances are directly used to construct the mini-batch online, thus, the mini-batch-based self-paced selection would leads to the mini-batch size changing during the entire learning procedure. To promise a fixed mini-batch size, the $K$ ousteds are replaced by the other samples randomly selected from training set and uniformly assigned the weights in $[0, 1]$. It is inspired by the tricks in Sachan & Xing (2016).

**Knowledge engineering.** In DA experiment, we specify two kind of structured knowledge based on Concept (Eq.38) and BP function (Eq.39). As what we have mentioned, the former indicates diversity where the subset are supposed to be selected to achieve sparsity across different conceptual groups $\{V_j\}_j^J$. The latter also consider data diversity (the first term in Eq.39), yet simultaneously perceive the complementarity (the second term in Eq.39) when the subset selection is performed. It means that, each group of $\{W_i\}_{i=1}^I$ are tend to be selected together.

Compared with the Concepts constructed in classification, the Concepts in DA are engineered within different manner. Specifically for source domain, $J$ concepts in DA are branched into two kinds. The first kind is based on categories. So there are 10 groups of concept (each concept refers to a specific class in the dataset). The second kind is constructed within a group of samples that belongs to the same category. We perform a similar feature-level clustering in each class and choose 5 clusters. Then totally we have $J = 10 + 5 * 10 = 60$ to construct the Concept structured knowledge in DA.

Now we turn to introduce how to construct BP function in DA. For a ablation study, the first term indicating diversity in BP function is completely the same of the Concept we previously mentioned. Then for the second term, we tend to choose the ambiguous examples into the same group. We train a prophet network then extract the classification activations across all training data, thus, use them to calculate the ambiguous semantic among training data. Then for each class, we sample a representative (the example closet to the centric) and collect 9 most similar examples from the other classes, respectively. So we have $I = 10$ in Eq.39 and each group contains 10 examples.

We apply the same routine to engineer the structured knowledge in the target domain.

SEQUENCE LEARNING.

DECIMAL NUMBER CALCULATION.

This task belongs to human-imitation task from AI perspective and is well-known that requires CCL methods to boost learning speed. In the basic setting, a sequence-to-sequence model (mostly, RNN) is employed to receive two-decimal-coded numbers separated by the sign "+", then produce the sum of those numbers in decimal coding (we only consider plus operation in this paper.). The evaluation is the minimum cost time to obtain the test accuracy more than 99%.

Our setup is similar to Zaremba & Sutskever (2014), yet the implementation is based on PyTorch. In our sequence-to-sequence models, the encoder and decoder are LSTMs with the same architecture containing 128 units. During training, all the CL baselines learn on 40,960 samples. Validation set consists of 4,096 examples and our batch size is set the same number.

**Curricula specification.** Decimal number addition includes a manual curriculum and ACL algorithm. The former is proposed based on the demonstration in Matiisen et al. (2017), ACL is directly applied to screen and select multiple tasks.

**Knowledge engineering.** We insist on diversity knowledge prior term *Concept*, while under the different knowledge engineering manner. Detailedly, we employ Hamming distance as the measure between digit examples then assign them into $J = 64$ clusters.

STOCK PRICE PREDICTION.

Our stock price data come from the daily data of the Chinese A share markets from July, 2007 to July, 2017. It is offered by a stock data service provider (www.wind.com.cn). The daily data for each stock contains six indexes including the Open Price, the Highest Price, the Lowest Price, the Close Price, the Turnover Rate and Volume. For our empirical study, we only use the Highest Price and Lowest Price for regression. MIN-MAX Normalized technique is used to promise all data value stay in the same order of magnitude.

We employ the opportunity window technique in Appel (2005), then 9423 windows of interest (at weekly granularity) are extracted by this strategy. Then LSTMs taught by different teachers are feed these time series at the daily granularity to predict the price. Our LSTM learner also consider multi-scale information, thus, the pricing information of the past 5, 10 and 20 days are encoded as a concatenated feature for regression.

**Curricula specification.** Different from decimal number addition, we have no information about which day or stock is easy for a LSTM model to learn. We use ACL as the only curriculum strategy to evaluate our T2T.

**Knowledge engineering.** The knowledge prior in stock price data should imply causality. In specific, we consider the prior as the set function in Eq.50 . Then all the opportunity windows from a specific stock, naturally maintain the time-based dependencies. We select the series within three months so

that detrend the possible longterm information that are not relevant for our daily analysis. Then these time-based aligned information constructs the target processes $Y$ in Eq.50 .

During implementation, we employ a trick to the casual subset selection. As can be observed, the samples not in the range of a specific time series are totally without causality. So instead of considering over all training data, we tend to consider each causality-based process respectively, then select $k$ training instances in total. Concretely, suppose we have $N_s$ aligned processes, then we separately select $\frac{k}{N_s}$ training instances within each process.

**Future price regression.** We perform the price regression task where the $9423$ windows of interest are trained to predict the lowest and highest prices within the next two weeks. The outcomes from ACL and ACL+SDK are demonstrated in Table.5 . As can be observed, ACL+SDK generally outperforms ACL, which also explains the higher accumulated profit in Fig.5 .

Table 5: The stock price regression results of ACL and ACL+SDK.

| | | ACL | | | | ACL+SDK | | | |
|---|---|---|---|---|---|---|---|---|---|
| | Errors | $0-10\%$ | $10-20\%$ | $20-40\%$ | $>40\%$ | $0-10\%$ | $10-20\%$ | $20-40\%$ | $>40\%$ |
| Price (Yuan) | | | | | | | | | |
| | $0-5$ | $0\%$ | $0\%$ | $0.47\%$ | $99.53\%$ | $0\%$ | $0\%$ | $0.56\%$ | $99.44\%$ |
| | $5-10$ | $1.26\%$ | $2.67\%$ | $16.96\%$ | $79.11\%$ | $5.98\%$ | $9.32\%$ | $25.47\%$ | $69.23\%$ |
| Highest | $10-15$ | $32.27\%$ | $28.68\%$ | $29.34\%$ | $9.7\%$ | $31.26\%$ | $31.29\%$ | $25.95\%$ | $8.7\%$ |
| | $15-20$ | $20.02\%$ | $29.66\%$ | $48.82\%$ | $1.49\%$ | $21.52\%$ | $34.16\%$ | $41.82\%$ | $2.5\%$ |
| | $>20$ | $4.05\%$ | $4.05\%$ | $32.83\%$ | $58.08\%$ | $7.5\%$ | $8.5\%$ | $34.38\%$ | $50.62\%$ |
| | $0-5$ | $0\%$ | $0\%$ | $0\%$ | $100\%$ | $0\%$ | $0\%$ | $0\%$ | $100\%$ |
| | $5-10$ | $1.67\%$ | $4.73\%$ | $16.48\%$ | $77.12\%$ | $1.67\%$ | $4.12\%$ | $18.76\%$ | $75.45\%$ |
| Lowest | $10-15$ | $31.24\%$ | $35.79\%$ | $21.65\%$ | $11.32\%$ | $28.43\%$ | $36.69\%$ | $26.75\%$ | $8.13\%$ |
| | $15-20$ | $22.6\%$ | $27.59\%$ | $42.40\%$ | $7.4\%$ | $10.05\%$ | $34.95\%$ | $46.44\%$ | $7.56\%$ |
| | $>20$ | $4.01\%$ | $6.09\%$ | $20.65\%$ | $69.25\%$ | $5.05\%$ | $5.05\%$ | $25.15\%$ | $64.75\%$ |

