# OpenReview forum: "Teaching to Teach by Structured Dark Knowledge"
_ICLR.cc/2019/Conference_

### Official Review · AnonReviewer2 · 2018-11-02
**General framework, insufficient comparison with prior work**

**Rating:** 6
**Confidence:** 5

**Review:**

Summarization: This paper studies how to inject structured prior knowledge into the teaching model for machine learning. The authors propose a very general framework called `teach to teach’, in which 1) the knowledge is distilled via subset selection that matches the teacher decision distribution 2) the distilled knowledge is then transferred into the teacher model by reweighting the contributions of teacher objective and coherence constraints. Extensive experiments are conducted on image classification, unsupervised domain adaptation and sequence learning.

Questions: 1) Can the teacher models, like those in L2T, be successfully transferred? For example, the teacher model trained with task 1 (with prior knowledge 1) successfully applied to task 2?
2) I’m not that clear about the relationship with knowledge distillation (Hinton et. al 2015). Per my understanding, the authors seem to make the distribution of the knowledge (specified by the set prior function F) coherent with teacher model and let the both influence each other (in section 3.2). In that sense I do not know what is the `dark’ knowledge here.

Pros: In general I think this paper is a decent work that the structural prior knowledge is elegantly combined with teaching strategy (a.k.a. the teacher models in curriculum learning). The proposed method is intuitive and natural. The empirical verifications are deep and comprehensive to demonstrate the effectiveness of the `teaching to teach’ framework.

Cons: 1) I think the authors should compare with self-paced learning with diversity (SPLD) since you also take diversity as a form of structural knowledge.
2) The writing needs to be significantly polished. First, please simply the writing both in terms of general logic and language. I spent quite a few efforts in figuring out the meaning of some notations and complicated terms such as `curriculum-routed’ and `g_i’. Furthermore, I see no reason of putting so much fancy decorations on an essentially iterative algorithm (the bottom part of page 5 and all page 6). Second,  I suggest the authors give more intuitive and concrete examples towards what is the structural prior knowledge at the earlier phase of the paper, rather than putting most of them into appendix. Last but not least, please use more clear citation formats: currently quite a few citations are missing of publishing venues such as Fan et.al 2018 and Furlanello et.al 2018.

---

### Official Review · AnonReviewer3 · 2018-11-05
**Major presentation issues**

**Rating:** 3
**Confidence:** 4

**Review:**

In this paper, the authors propose a new technique called Teaching to Teach via Structured Dark Knowledge for curriculum learning.  See my comments below.

I don’t like this paper because it is full of buzzwords, and it is really poorly written. The author first started in the abstract on “hyper deep learners”, which confuses me a lot. It is unclear to me what type of deep learning models that you are aiming for and what exactly “hyper deep learners” are. Also, in the abstract, the authors mention Structured Dark Knowledge, but they have not discussed it in the introduction, which makes it extremely hard for the readers to understanding the relationship between this work and the so called “Structured Dark Knowledge”.

The 3.2 is also poorly written and insufficiently motivated. The term Structured Dark Knowledge sounds fancy, but I fail to see what the structures are. It sounds like the authors just propose “Training Subset” as Structured Dark Knowledge, which is extremely misleading. This work gives people an impression that you are trying to leverage external knowledge for curriculum learning, it turns out it is not the case.

In section 4.1, the hyperparameters’ setting seems to be mysterious. It is unclear to me how the authors come up with magic numbers like 0.04, 0.1, 15%.

In Figure 2, the improvements from SDK do not look like it is very impressive. And also 4.2 the datasets are too small. The metric in Table 4 is poorly chosen. I don’t understand what the numbers mean un Table 4 and how significant they are. Please use standard metrics.

The future stock price regression experiments are also poorly presented. In Table 5, the authors do not explain what the price, percentage and error mean. It only says “Results”. Please pick common benchmark datasets and well-known metrics.

Overall, this paper is poorly written, and it has not met the requirement of ICLR.

---

### Official Review · AnonReviewer4 · 2018-11-13

**Rating:** 4
**Confidence:** 1

**Review:**

Summary
==============
This paper proposes "teaching to teach". A drawback of existing curriculum learning approaches is that it requires a human to manually and heuristically define a curriculum. Teaching to teach avoids this using a teacher model to perform subset selection, and reweighting the data points based on the teacher loss.

Evaluation
==============
Unfortunately, after reading section 3 a few times I am still not 100% clear on the exact methodology. It could be lack of background on my part, but I find the presentation extremely confusing, and unnecessarily verbose/florid. Also I am not exactly sure how the proposed methodology relates to Hinton et al's "knowledge distillation", which trains a (typically smaller) student model to mimic the output from a teacher model. The empirical study seems to be strong however, with applications across various domains/datasets and sensible baselines.

---

### Meta-Review · Area_Chair1 · 2018-12-14
**Lacking clarify; no rebuttal submitted**

**Confidence:** 5
**Recommendation:** Reject

**Metareview:**

This work proposes a "teaching to teach" (T2T) method to incorporate structured prior knowledge into the teaching model for machine learning tasks. This is an interesting and timely topic. Unfortunately, among many other issues, this paper is fairly poorly writing and can benefit from a significant rewriting. The authors did not provide a rebuttal and hence we recommend a rejection.